# Spin of fractional quantum Hall neutral modes
# and "missing states" on a sphere

Dung Xuan Nguyen[1,2], and Dam Thanh Son [3]

**1** Center for Theoretical Physics of Complex Systems, Institute for Basic Science, (IBS), 34126 Daejeon, Korea
**2** Basic Science Program, Korea University of Science and Technology (UST), 34113 Daejeon, Korea
**3** Kadanoff Center for Theoretical Physics, University of Chicago, 933 E 56th St, Chicago, Illinois 60637, USA

March 2025

## Abstract

A low-energy neutral quasiparticle in a fractional quantum Hall system appears in the latter's energy spectrum on a sphere as a series of many-body excited states labeled by the angular momentum $L$ and whose energy is a smooth function of $L$ in the limit of large sphere radius. We argue that the signature of a nonvanishing spin (intrinsic angular momentum) $s$ of the quasiparticle is the absence, in this series, of states with total angular momentum less than $s$. We reinterpret the missing of certain states, observed in an exact-diagonalization calculation of the spectrum of the $\nu = 7/3$ FQH state in a wide quantum well as well as in many proposed wave functions for the excited states as a consequence of the spin-2 nature of the zero-momentum magnetoroton.

## 1 Introduction

Recently, the existence of spin-2, electrically neutral, degrees of freedom in fractional quantum Hall (FQH) systems has become a topic of considerable interest, both theoretically and experimentally[1]. The first theoretical suggestion of a spin-2 mode was made in 2011 by Haldane [1], who proposed that FQH systems possess a collective degree of freedom in the form of an unimodular 2D spatial metric. Later, Maciejko et al. [2] proposed that such a mode would appear near a nematic phase transition and identified it with the long-wavelength magnetoroton [3]. Golkar, Nguyen, and Son [4] showed how the spin-2 mode fits in the framework of sum rules. There is now solid numerical evidence for the existence of the spin-2 mode [5,6], including evidence for the existence of multiple spin-2 modes in the Jain sequence near $\nu = 1/4$ [7–9]. Most recently, following the suggestions of Refs. [4,6,10], the spin-2 mode has been observed using circularly polarized Raman scattering [11]. Although the rotational symmetry is broken to $C_4$ by the lattice, the effect of this breaking is numerically small [10], which allows the experiment to determine the sign of the projection of the spin onto the magnetic field direction. The experimental result of Ref. [11] is consistent with theoretical expectation for $\nu = 1/3, 2/3, 2/5$, and $3/5$.

In this short note, we argue that the fact that the long-wavelength magnetoroton has a spin equal to 2 has a distinct consequence for the spectrum of the FQH system on a sphere. In fact, we argue that evidence for the spin-2 nature of the long-wavelength magnetoroton has been long

---

[1]By the "spin" of a quasiparticle one has in mind its angular momentum when it linear momentum is zero. At the microscopic level, the spin comes entirely from the orbital motion of the electrons; we assume that the electrons are fully spin-polarized.

hidden in numerical results of Ref. [12], as well as in various constructions of the excited-state wave function (see, e.g., Refs. [13–16]).

## 2 The puzzle of missing states

Consider a gapped FQH state. Let us assume that the system possesses a quasiparticle with a dispersion relation that corresponds to a gap $E_0$ at zero momentum and an effective mass $m$,

$$E(\mathbf{k}) = E_0 + \frac{k^2}{2m} + O(k^4). \tag{2.1}$$

In the FQH context, only electrically neutral particles can be assigned a dispersion relation. (Equation (2.1), thus, cannot be applied to, e.g., the Laughlin quasihole or quasielectron.) For simplicity, let us assume that the quasiparticle (2.1) has a much lower energy than any other excitations in the system, although much of the discussion that follows is valid under a much weaker condition.

Let us now put the system on a torus of size $L_x \times L_y$, where $L_x$, $L_y$ are much larger than the magnetic length $\ell_B$. The number of flux quanta going through the torus is large, $N_\phi = L_x L_y / (2\pi \ell_B^2) \gg 1$, so is the number of electrons, $N_e = \nu N_\phi \gg 1$. The FQH system has several degenerate ground sates; we consider one of these. The excited states with the lowest energy above this ground state contain a single quasiparticle with energy given by the same formula as Eq. (2.1), except now the momentum has discrete values,

$$E = E_0 + \frac{1}{2m} \left[ \left( \frac{2\pi n_x}{L_x} \right)^2 + \left( \frac{2\pi n_y}{L_y} \right)^2 \right], \tag{2.2}$$

where $n_x$ and $n_y$ are integers.

Now we put the system on a sphere. We will assume that in the ground state there is no charged excitations (e.g., quasihole or quasielectron). That requires the number of electrons to be related to the number of flux quanta threading the sphere as $N_e = \nu(N_\phi + S)$, where $S$ is the shift of the state. We assume that the ground state has zero angular momentum.

The excited states are organized in multiplets under the SO(3) rotational symmetry. If the excited state consists of a single quasiparticle, the energy of the excited state is simply the kinetic energy of a particle with mass $m$ and angular momentum $L$ moving on a sphere of radius $R$. Hence one expects

$$E(L) = E_0 + \frac{L(L+1)}{2mR^2}. \tag{2.3}$$

For concreteness, we focus on one particular quantum Hall system, namely, the $\nu = 1/3$ state. The lowest-energy neutral quasiparticle in this case is the magnetoroton [3]. For the $\nu = 1/3$ FQH state with Coulomb interaction the magnetoroton dispersion relation has a minimum at $k_0 \sim 1/\ell_B$ (hence the name) and a local maximum at $\mathbf{k} = 0$. The magnetoroton at $\mathbf{k} = 0$ is, however, buried in a continuum of states, which presumably can be understood as two-quasiparticle states containing two magnetorotons near the minimum of the dispersion curve. The long-wavelength ($\mathbf{k} \approx 0$) magnetoroton, thus, should be unstable against decay into two finite-momentum magnetorotons. The instability of the long-wavelength magnetoroton is, however, not an unavoidable consequence of the FQH effect. In fact, numerical simulation of the $\nu = 7/3$ state (i.e., $\nu = 1/3$ state on the second Landau level) in a wide quantum well [12] reveal a regime where the magnetoroton at $\mathbf{k} = 0$ is stable, with an energy less than twice the energy at the minimum at $\mathbf{k} \neq 0$, and with a positive effective mass. One expects that in this case the series of states (2.2) and (2.3) can be reproduced numerically.

Exact-diagonalization study conducted in Ref. [12] with the number of electrons ranging from 11 to 13 largely confirmed theoretical expectation, but with an unexpected twist. While on the torus one indeed finds a series of states parametrized by the discrete momenta $n_x$ and $n_y$ which seem to run over all integer values including $n_x = n_y = 0$, on a sphere one finds low-energy excited states with $L = 2, 3, 4, \ldots$, but not with $L = 1$ or $L = 0$. In Ref. [12] the absence of these two states was attributed to finite-size effects, with the expectation that they would reappear in simulations with a much larger system size.

## 3 The main argument

We argue here that the absence of the $L = 0$ and $L = 1$ modes actually persists to an arbitrarily large sphere radius, and that this is a signal of the spin-2 nature of the long-wavelength magnetoroton.

The main idea is the following. When a quasiparticle carrying a nonzero intrinsic angular momentum $s$ moves along a closed contour $\mathcal{C}$ on a sphere, it acquires a Berry phase due to the coupling of the intrinsic angular momentum with the spin connection. Indeed, the spin of the quasiparticle is directed perpendicularly to the surface of the sphere, so when the quasiparticle moves, the direction of its spin changes. Thus the quasiparticles acquire a Berry phase

$$\gamma[\mathcal{C}] = s\Omega[\mathcal{C}], \tag{3.1}$$

where $\Omega[\mathcal{C}]$ is the solid angle subtended by $\mathcal{C}$. This Berry phase is the result of the coupling of the spinful particle to the spin connection on the sphere[2].

The Berry phase makes the problem equivalent to that of a particle moving on a sphere under the influence of the magnetic field of a magnetic monopole located at the center of the sphere. We emphasize that this magnetic monopole is fictitious and unrelated to the source of the magnetic field that gives rise to the FQHE. The quasiparticle under consideration is charge-neutral with respect to the physical electromagnetic field, but its spin makes it "feels" a fictitious magnetic field.

The problem of a particle of charge $e$ moving a sphere of radius $R$ in the field of a magnetic monopole of magnetic charge $g$ was solved by Wu and Yang [17] (see also Ref. [18]). The energy levels are given by

$$E = \frac{L(L+1) - (eg)^2}{2mR^2}, \tag{3.2}$$

where $L$ is the angular momentum. It can be demonstrated that $E \geq 0$[3]. Consequently, unlike the situation without a magnetic monopole, here $L$ cannot take arbitrary integer values, but only

$$L = |eg|, \ |eg| + 1, \ |eg| + 2, \ldots \tag{3.3}$$

On the other hand, the charge of the fictitious magnetic monopole that gives rise to the Berry phase (3.1) is

$$g = \frac{s}{e}. \tag{3.4}$$

That means that the allowed values for the angular momentum on the sphere are

$$L = |s|, \ |s| + 1, \ |s| + 2, \ldots \tag{3.5}$$

---

[2]We include a discussion of this coupling in the Appendix.

[3]For the detailed derivation, see the Appendix.

States with $L < |s|$ do not appear in the spectrum. Thus, the absence of $L = 0$ and $L = 1$ modes (and the presence of $L \geq 2$ modes) observed in Ref. [12] is not a finite-size effect, but a direct evidence showing that spin of the long-wavelength magnetoroton is 2 (more precisely, $+2$ or $-2$).

Interestingly, the constraint (3.5) is quite general for other gapped neutral excitations in fractional quantum Hall systems. In non-Abelian FQH states, neutral fermion modes do exist—particularly in the Moore-Read and Haldane-Rezayi states, where there is a neutral fermion excitation carrying spin $3/2$. In fact, exact diagonalization on a sphere reveals that the neutral fermion eigenstate begins at the angular momentum $L = 3/2$, while the eigenstate with $L = 1/2$ is absent from the spectrum [19, 20].

In the bilayer bosonic quantum Hall system with the filling fraction $\nu_T = 1/3 + 1/3$, there is a chiral spin-1 mode that corresponds to the interlayer dipole moment [21]. From our argument above, the corresponding angular momentum $L = 0$ should be missing in the energy spectrum on a sphere. Indeed, numerical results confirmed its absence [21].

# 4 Conclusion

We note that before Ref. [12], the lack of the $L = 1$ excited state has already been encountered in many studies. In particular, in the composite-fermion framework, the magnetoroton is obtained by moving a composite fermion from a lower, occupied $\Lambda$ level to a higher, unoccupied one, and then projecting the resulting wavefunction to the lowest Landau level. In the $\nu = 1/3$ case, one can create a state with $L = 1$ by promoting a composite fermion from the lowest $\Lambda$-level to the next $\Lambda$-level, but that state annihilated, in a nontrivial manner, by the projection to the lowest Landau level [13, 14]. The absence of the $L = 1$ mode is also a feature of analytic constructions of the excited states directly on the lowest Landau level [15, 16]

It should be mentioned that the *sign* of the spin of the long-wavelength magnetoroton cannot be determined from the argument that we have presented. In particular, the $\nu = 1/3$ and $\nu = 2/3$ cases have exactly the same missing states in the sphere spectrum.

It has been argued that Jain states with $\nu = n/(2n \pm 1)$ accommodate not only spin-2 but also spin-3, spin-4, etc., excitations [22, 23]. If a spin-$s$ mode is a well-defined excitation, one should observe the missing of corresponding modes with $L < s$ on the sphere. This may be related to the "exclusion rules" for counting excited states in the composite fermion picture [14].

Recent numerical studies have found evidence of the magneto-roton in fractional Chern insulator (FCI) Laughlin states, using the stress tensor operator [24, 25]. In particular, by tuning the interaction in the FCI system of twisted MoTe$_2$, the gap of the spin-2 mode at the zero momentum can be lowered[4], making the spin-2 mode a stable excitation [25].

# Acknowledgements

The authors thank Ajit Balram, Duncan Haldane, Janendra Jain, Thierry Jolicoeur, Edward Rezayi, and Kun Yang for discussions. We also thank Ajit Balram and Bo Yang for commenting on the previous version of our manuscript. The work of DTS is supported, in part, by the U.S. DOE grant No. DE-FG02-13ER41958 and by the Simons Collaboration on Ultra-Quantum Matter, which is a grant from the Simons Foundation (No. 651442, DTS). DXN is supported by the Institute for Basic Science in Korea through Project IBS-R024-D1.

---

[4]Similarly with the proposal for FQHE in Ref. [12].

# A  Energy spectrum of neutral modes on a sphere

For completeness, in this Appendix we provide a detailed derivation of Eqs. (3.2) and (3.3).

## A.1  Metric and notation

We consider the 2-sphere with radius $R$ with the metric $g_{ij}$ $(i, j = \theta, \phi)$ in the spherical coordinate

$$ds^2 = g_{\theta\theta}(\theta, \phi)d\theta^2 + g_{\phi\phi}(\theta, \phi)d\phi^2, \tag{A.1}$$

where the metric components are explicitly given by

$$g_{\theta\theta}(\theta, \phi) = R^2, \quad g_{\phi\phi} = R^2 \sin^2 \theta. \tag{A.2}$$

The spatial vielbein $e_i^a$ is given as

$$e_\theta^1 = R, \quad e_\phi^1 = 0, \quad e_\theta^2 = 0, \quad e_\phi^2 = R \sin \theta, \tag{A.3}$$

which satisfies the relation to the metric $g_{ij} = \delta_{ab} e_i^a e_j^b$. The spatial spin connection is given by [26]

$$\omega_i = \frac{1}{2} \left( \epsilon^{ab} e^{aj} \partial_i e_j^b - \varepsilon^{jk} \partial_j g_{ki} \right), \tag{A.4}$$

where $\varepsilon^{ij} = \epsilon^{ij}/\sqrt{g}$ defined through the anti-symmetric tensor $\epsilon^{12} = -\epsilon^{21} = 1$ and the determinant of the metric $g = \det(g_{ij}) = R^4 \sin^2 \theta$. We now can compute the components of the spatial spin connection:

$$\omega_\theta = 0, \quad \omega_\phi = -\cos \theta. \tag{A.5}$$

Note that since we are working on a static curved space in $2 + 1D$, the spin connection defined in equations (A.4) and (A.5) correspond to $\omega_i^{12} = -\omega_i^{21}$ in the usual notation of the general geometry [27, 28]. The spin connection (A.5) also satisfies the relation to the Ricci curvature of a sphere with radius $R$

$$\mathcal{R} = \frac{2}{\sqrt{g}} \left( \partial_\theta \omega_\phi - \partial_\phi \omega_\theta \right) = \frac{2}{R^2}. \tag{A.6}$$

From the above convention, on a static curved space, the covariant derivative acting on the neutral field $\Psi$ with the *spin*$-s$ representation of rotational group $SU(2)$ is given by

$$D_i \Psi = \left( \partial_i + \frac{i}{2} \omega_i^{ab} \Sigma_{ab} \right) \Psi, \quad (i = \theta, \phi). \tag{A.7}$$

$\Sigma_{ab}$ is the component of spin operator in spin-$s$ representation in the direction perpendicular to the plane $(a, b)$. Since we have the only non-zero component of spin connection in the static curved space in $2 + 1D$ are $\omega_i^{12} = -\omega_i^{21} = \omega_i$, only $\Sigma_{12} = -\Sigma_{21}$ appear in the covariant derivative (A.7).

## A.2  Energy spectrum of a spin-$s$ excitation on a sphere

We consider the neutral excitation to have a spin $s$ that can be oriented either inward or outward relative to the sphere—corresponding to negative or positive chirality, respectively. Consequently, we obtain

$$\Sigma_{12}\Psi = s\Psi, \tag{A.8}$$

where $\Psi$ represents the spin-$s$ field exhibiting either positive or negative chirality. In this expression, the $s > 0$ denotes an outward-pointing spin, while the $s < 0$ indicates an inward-pointing

spin relative to the sphere. With the assumed dispersion relation on a flat space (2.1). The action of the neutral excitation on a sphere is now given

$$
S = \int dt d\theta d\phi \sqrt{g} \left\{ i\Psi^\dagger \partial_t \Psi - \Delta \Psi^\dagger \Psi - \frac{g^{ij}}{2m} \left( \partial_i - is\omega_i \right) \Psi^\dagger \left( \partial_j + is\omega_j \right) \Psi \right\}, \tag{A.9}
$$

For the excitations of FQH systems, we explicitly need to consider that both the energy gap $\Delta$ and the effective mass $m$ are functions of the ratio $\ell_B/R$, where $\ell_B$ denotes the magnetic length and $R$ represents the radius of the sphere. The functions $\Delta \left( \ell_B/R \right)$ and $m \left( \ell_B/R \right)$ depend on the specific FQH state and the nature of the considered excitation. We apply a gauge transformation for the north hemisphere and the south hemisphere

$$
\Psi_N = \Psi e^{-si\phi}, \quad \Psi_S = \Psi e^{si\phi}. \tag{A.10}
$$

The Schrödinger equation is obtained as the field equation of the action (A.9),

$$
i\partial_t \Psi_{N/S} = -\Delta \Psi_{N/S} - H_{N/S} \Psi_{N/S}. \tag{A.11}
$$

With $H_N$ is the Hamiltonian of a non-relativistic particle with charge $e$ on a sphere with a magnetic monopole $g$ put at the center of the sphere, such that $eg = s$ [17],

$$
H_g^N = \frac{1}{2m} \left( -i\vec{\nabla}_s - e\vec{A}^N \right)^2, \tag{A.12}
$$

with $\vec{\nabla}_s$ is the gradient derivative projected to the sphere. The gauge potential induced by the magnetic monopole is given by

$$
A_\phi^N = g \frac{1 - \cos\theta}{R \sin\theta}. \tag{A.13}
$$

Similarly, the south hemisphere Hamiltonian is

$$
H_S = H_g^S = \frac{1}{2m} \left( -i\vec{\nabla}_s - e\vec{A}^S \right)^2, \quad A_\phi^S = -g \frac{1 + \cos\theta}{R \sin\theta}. \tag{A.14}
$$

The gauge choice on the north and south hemispheres is a convention so that the gauge field is non-singular at the north pole ($\theta \to 0$) or south pole ($\theta \to \pi$).

From now on, we choose the north hemisphere convention, and skip the subscript index $N$ for simplicity, and replace $H_N$ with $H$. We define the following angular momentum operator,

$$
\vec{J} = \vec{r} \times \left( -i\vec{\nabla}_s - e\vec{A} \right) - eg\hat{r}, \tag{A.15}
$$

which behaves as the usual angular momentum operator with the commutation relation $[J_i, J_j] = i\epsilon^{ijk} J_k$. We can check explicitly that the Hamiltonian is rotational invariant $[J_i, H] = 0$. Therefore eigenstates of the Hamiltonian $H$ can be chosen to have well-defined total angular momentum,

$$
J^2 |\Psi_L\rangle = L(L+1)|\Psi_L\rangle. \tag{A.16}
$$

One can explicitly show that

$$
H = \frac{1}{2m} \left( -i\vec{\nabla}_s - e\vec{A} \right)^2 = \frac{1}{2mR^2} \left( J^2 - e^2 g^2 \right). \tag{A.17}
$$

Thus the eigenstate of $H$ with total angular momentum $L$ has the eigenenergy

$$
E_L = \frac{L(L+1) - (eg)^2}{2mR^2}, \tag{A.18}
$$

which is Eq. (3.2) in the main text. Furtheremore, since the Hamiltonian is rotational invariant, the degeneracy of each energy level is $2L + 1$. Since the momentum operator $\vec{\Pi} = -i\vec{\nabla}_s - e\vec{A}$ is Hermitian, it is obviously that

$$\langle \Psi_L | \left( -i\vec{\nabla}_s - e\vec{A} \right)^2 | \Psi_L \rangle \geq 0, \tag{A.19}$$

which gives us the constraint of the angular momentum of an eigenstate

$$L(L+1) \geq e^2 g^2 = s^2. \tag{A.20}$$

Then Eq. (3.3) is immediately obtained. For the magnetoroton mode with spin equal to 2, $L \geq 2$ was found in exact diagonalization studies.

### A.3  Other neutral excitations

#### A.3.1  Neutral fermionic modes of the Moore-Read and Haldane-Rezayi states

References [15, 19] suggest the existence of spin-$3/2$ excitations which are the neutral fermionic modes in the Moore-Read and Haldane-Rezayi states. When one computes the spectrum on a sphere, the angular momentum of the eigenstates should satisfy the constraint

$$L(L+1) \geq \left( \frac{3}{2} \right)^2 = \frac{9}{4}. \tag{A.21}$$

Consequently, the eigenstate with angular momentum $L = 1/2$ is absent. The same conclusion was suggested in Refs. [15, 19] by the trial wavefunction construction.

#### A.3.2  Interlayer-dipole excitations of bosonic bilayer $\nu_T = \frac{1}{3} + \frac{1}{3}$

In the system of bosonic bilayer quantum Hall with filling fraction $\nu_T = \frac{1}{3} + \frac{1}{3}$, the authors of Ref. [21] suggested there is a chiral spin-1 mode which is an interlayer-dipole excitation. We suggest the effective theory for inter-dipole excitations has the form

$$\mathcal{L} \sim i\bar{P}\partial_t P + \Delta \bar{P}P - \frac{1}{2m^*}|\partial_i P|^2 + E_i^- P^i, \tag{A.22}$$

with $P^i$ as the interlayer-dipole moment, $E_i^-$ is the difference between electric fields applied to the layers, and

$$P = P_x + iP_y, \quad \bar{P} = P_x - iP_y. \tag{A.23}$$

In the action (A.22), both $\Delta$ and $m^*$ depend on the interlayer distance $d$. In the absence of the applied electric field $E_i^-$, one can repeat the calculation in and find the constraint on the angular momentum of the dipole excitations excitation on a sphere

$$L(L+1) \geq 1. \tag{A.24}$$

Consequently, the eigenstate with angular momentum $L = 0$ is absent from the energy spectrum obtained via exact diagonalization on a spherical geometry.

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
