# Peer review of "Spin of fractional quantum Hall neutral modes and "missing states" on a sphere"

_SciPost Physics_

## Round 1 · Referee Report · Anonymous (Referee 1) · 2025-6-5

Strengths
Weaknesses
I am also not convinced by the claim that higher spin excitations in nu=n/(2n+/-1) Jain states should start from increasingly higher values of L. Such modes have been studied in CF theory framework, e.g., in https://www.nature.com/articles/nphys1275 but it does not seem that any states are missing (apart from L=1). How does this numerical observation fit in with the authors' arguments?
Another general question is what happens if one considers a different type of geometry, such as a (finite) cylinder or disk. One should be able to directly map states between these geometries and the sphere (e.g., stereographically), but it is less clear to me if the authors' argument would work in these cases.
Report
Recommendation
Ask for major revision
In this brief note, the authors provide an a posteriori explanation for the angular momentum structure of collective modes in fractional quantum Hall states, namely the fact that the dispersions of these modes often start from a well-defined, non-zero value of angular momentum. The argument is based on the Berry phase contribution of a fictitious magnetic monopole that is due to the intrinsic spin of the excitation. Overall, the argument looks plausible and it seems to account well for a few well-known cases in the literature.
We thank the referee for carefully reading our manuscript and providing constructive comments.
Q1.1 However, the manuscript feels a bit "thin" on the content and I am not sure if it tells the whole story. For example, the role of LLL projection is not very transparent. The absence of L=1 state is not limited to the Laughlin or composite fermion states -- this state vanishes simply due to the fact that the ground state is a singlet (L=0) and the LLL-projected density operator essentially acts as a raising operator (see https://journals.aps.org/prb/pdf/10.1103/PhysRevB.50.1823). Based on this argument, I would expect that if we relax the condition of LLL projection, we can have situations where L=1 excitation becomes possible.
A1.1 The referee’s argument is based on Girvin, MacDonald, and Platzman’s variational ansatz for the magneto-roton. In contrast, our conclusion is general; it does not depend on the microscopic construction of the excitation. Moreover, our results can be applied to more general systems and more general neutral excitations, not only the magneto-roton in the single-layer fractional quantum Hall systems. For example, in the case of the quantum Hall bilayer, our argument implies the absence of the L=0 spherical mode of the neutral spin-1 excitation. It is correct that our conclusion for the magneto-roton excitation of the FQH system assumed the lowest Landau level constraint from the beginning. It is also accurate that if the lowest Landau level constraint is released, then we expect the excitation with L=1 to become possible: it is the Kohn mode mentioned by the second referee (Prof Steve Simon).
Q1.2 For example, what would happen if one took nu=2/5 Jain state in the lowest two Landau levels? With the condition that the two LLs are degenerate in energy, this state has the exact parent Hamiltonian (V1 pseudopotential) and, presumably, a well-defined collective mode exists. What do the authors predict for such a case, would the collective mode start from L=1 or not?
A1.2 See our response to the referee’s question on the general Jain states and to Question Q2.4 raised by the second referee.
Q1.3 Similarly, one could ask what they predict for higher order Read-Rezayi states?
A1.3 The referee raises interesting questions, but they are outside the scope of the current paper. We didn’t predict the absence of L=1 for higher-order Read-Rezayi states. On the other hand, for the case of the neutral excitation in Read-Rezayi state, Sreejith et al. [Phys. Rev. B 87, 245125 (2013)] predicted that the angular momentum begins at L=2.
Q1.4 I am also not convinced by the claim that higher spin excitations in $\nu=n/(2n\pm 1)$ Jain states should start from increasingly higher values of L. Such modes have been studied in CF theory framework, e.g., in https://www.nature.com/articles/nphys1275 but it does not seem that any states are missing (apart from L=1). How does this numerical observation fit in with the authors' arguments?
A1.4 It is correct that in the paper by Majumder, Mandal, and Jain mentioned by the referee [https://www.nature.com/articles/nphys1275] only the L=0 and L=1 states are found to be forbidden. However, we need to take a closer look at the counting of the angular momentum of the neutral excitation on a sphere. In a paper by Balram, Wójs, and Jain [Phys. Rev. B 88, 205312 (2013)] (Ref [14] in our manuscript), one can see that there are fewer states with L=2 than with L=3. More generally, the number of states increases with L. This can be understood if one assumes that states with a given L may come from different neutral quasiparticles with different spins, such that a spin-s neutral quasiparticle gives rise to a branch of excited states with angular momentum starting from L=s. Please refer also to our responses to Questions Q2.4 and Q2.5 from the second referee for further details.
Q1.5 Another general question is what happens if one considers a different type of geometry, such as a (finite) cylinder or disk. One should be able to directly map states between these geometries and the sphere (e.g., stereographically), but it is less clear to me if the authors' argument would work in these cases.
A1.5 The punchline of our paper is that the “missing states” on a sphere, compared to on the torus, give us the spin of the gapped neutral excitations. We use the effective field theory argument rather than the previous constructions using the wave function or the composite fermion. The advantage of our argument is that we can apply it to general gapped neutral excitations on different physical systems. We do not intend to map the states from spherical geometry to the torus geometry.

Author: Dung Nguyen on 2025-09-16 [id 5826]
(in reply to Report 3 on 2025-06-26)This is a succinct, clear, and beautiful paper, providing a new and very interesting argument for the missing states at some angular momenta in neutral excitation modes of fractional quantum Hall (FQH) systems. While the "missing states" has been known before, this paper interprets it from a novel aspect based on Berry phase. I like the Berry-phase based analogy between a neutral quasiparticle with intrinsic angular momentum on a sphere and a charged particle on a sphere with a fictitious magnetic monopole at the center. I would recommend the publication in SciPost Physics, if my following comments can be properly addressed. The other two referees also raised important questions.
We thank the referee for carefully reading our manuscript and giving important comments. We will address the comments in detail in the following.
Q3.1 (*) The dispersion of the neutral excitation is assumed to be quadratic in Eq. (2.1). To what extent does the argument in Sec. 3 depend on this assumption? Is it possible that in the long wavelength limit some FQH neutral excitation modes do not obey the quadratic dispersion with k=0 as a maximum/minimum? I suggest the authors to discuss this issue.
A3.1 We thank the referee for raising this interesting point. Here we consider the rotational symmetry; therefore, the dispersion should have the form $E(k)=\Delta+ c_2 k^2 +c_4 k^4+\cdots$. The main conclusion in section 3 doesn’t depend explicitly on the dispersion, given that the Hamiltonian is invariant under rotational symmetry discussed in Appendix A ($[H,J_i]=0$), such that the eigenstates have a well-defined total angular momentum. The main conclusion comes from equation (A.19), which is true for all eigenstates of the effective Hamiltonian with rotational symmetry, and (A.20) logically follows from (A.19). We added a paragraph in Appendix A to address this comment by the referee.
Q3.2 By the way, the effective mass in Eq. (2.1) can be negative, right? At least the magnetoroton mode of the $\nu=1/3$ Laughlin state in the lowest Landau level goes up when k->0, suggesting a negative mass.
A3.2 The mass is indeed negative for $\nu= 1 / 3$. We used the positive mass, relevant for the case of $\nu= 7/3$ considered by Jolicoeur [Phys. Rev. B 95, 075201 (2017)]. In the case of negative mass, the main conclusion on the “missing states” remains the same.
We replaced “It can be demonstrated that $E \geq 0$ ” to “It can be demonstrated that $L(L+1)-(eg)^2 \geq 0$”. We also further added a foot note [2] to discuss the explicit form of the dispersion. We also add a paragraph under equation (A.20) explaining that the main conclusion doesn’t depend on the explicit choice of the dispersion relation, as long as the rotational symmetry is respected.
Q3.3 (*) In the last paragraph of Sec. 2, the authors mentioned the difference between the low-energy excitations on the torus and on the sphere. This reminds me one thing which I didn’t understand well. For the $\nu=1/3$ Laughlin state, the magnetoroton mode on the sphere starts from angular momentum L=2 -- no states at L=1 and L=0 in this mode. So the long wavelength limit of this mode, the so called spin-2 graviton, should be carried by the L=2 state for finite systems. As the “momentum” on the sphere is often defined as L/R, where R is the sphere’s radius, this graviton actually has a finite momentum for finite systems on the sphere. As the Laughlin ground state has L=0, the graviton on the sphere has different L and (for finite systems) different “momentum” from the ground state. By contrast, spin-2 spectral function calculations on the torus, like those done in Refs. [5] and [6], found evidence of the spin-2 graviton exactly in the momentum k=0 sector even for finite systems. As the ground state is also in the k=0 sector on the torus, this means the graviton and the Laughlin ground state already have the same momentum for finite systems on the torus. This is quite different from the situation on the sphere. Why does not the graviton appear at a finite momentum (nonzero nx or ny) on the torus for finite systems? Can the authors comment on this discrepancy between different geometries?
A3.3 The main goal of our paper is exactly to explain and highlight the difference between the sphere and the torus geometries. A neutral excitation with a nonzero “spin” couples to the background geometry not only through the metric but also through the spin connection. In 2D, the coupling of the spin of the excitation with the spin connection is similar to that of an electric charge with the gauge potential. In this mapping, the curvature of the background manifold corresponds to the magnetic field, and the “spin” corresponds to the electric charge. In the case of torus geometry (zero curvature), the “spin” plays no role, so the graviton can appear at zero momentum.
It is worth noting that on the torus, one cannot construct the k=0 magneto-roton excitation using the Girvin, MacDonald, and Platzmann’s method: the projected density operator at k=0 is proportional to the identity operator. However, one can generate the graviton mode at $k=0$ by using two-body operators (the chiral stress tensor operators; see, e.g., Phys. Rev. Lett. 123, 146801 (2019) and Phys. Rev. Lett. 128, 246402).

---

## Round 1 · Referee Report · Steven Simon (Referee 2) · 2025-6-12

Report
I have a few minor comments. I found some typos. And I have a few suggestions.
Let me start with the minor things:
(1) Suggestion. Just after eq. 2.1, it is worth saying a few more words about why only neutral particles have a dispersion. I had to think for a while to understand what you meant and why this is true.
(2) Change of language. I think about 7 lines up from the bottom of page 2, the word "should" would be better as "could", since in the very next sentence you say this is not always the case.
(3) Missing word. 6 lines below the start of the conclusion the word "is" is missing. (but that state IS annihilated)
Now a larger suggestion for discussion of some additional phyiscs
which I think would add to the paper nicely:
I think it is worth comparing your result, at least for the case of
the jain series, to the spectra expected from composite fermion
excitons. Maybe this should be done in an appendix, maybe in the
discussion section, or some combination. For nu=1/3, just by
constructing all possible excitons --- a single particle in an excited
CFLL and a single hole in the valence CF LL --- one obtains 1 state at L=1, 2 states at L=2, 3 states at L=3 and so forth in the thermodynamic limit. (This is trivial to show just by counting since the valence level has some angular momentum L0 and the first empty level has angular momentum L0+1 then next has L0+2 etc, and you want to add the hole in L0 to the electron in L0+M to generate some general L.) In the Jain picture, the L=1 exciton state vanishes under projection but it is unclear exactly why. In the HLR picture it is more clear (using some mass renormalized version of HLR --- see the reference from the other referee, thank you for the citation!). Here the Kohn mode is part of the spectrum, so it grabs one state at each L. So the low energy part of the spectrum then gets 0 states at L=1, 1 state at L=2, 2 states at L=3 and so forth. In your language would you say that we are seeing one spin-2 branch, one spin-3 branch and so forth? At nu=2/5, just by counting excitons again, you get 1 state at L=1, 3 states at L=2, 5 states at L=3 and so forth. Again one mode of each L is pushed up to the Kohn mode leaving 0 states at L=1, 2 states at L=2, 4 states at L=3 and so forth. So you would say that we have 2 modes of spin 2, 2 more modes of spin 3 and so forth? Is this correct? If so, it then suggests that one should go looking for these in numerics. However, one thing that seems special about the magnetoroton is that, at least under some circumstances, it doesn't get buried in the continuum. Is there any hope that these higher spin modes could also be out of the continuum and could be observed?
Because I was a bit slow in writing this report (just a few days!), I did have the benefit of reading one of the reports from another referee. That referee says that this work is "thin on results." I would actually disagree. While I would not object to extending this work or addressing some of the above issues (which overlaps with the issues raised by the other referee), I think the paper really does stand on its own. The paper makes a clear point and is an important point to make. It reminds me of the good old days of PRL (1950s) when a "letter" was often one page or less --- and these papers are very memorable. Now, much to our detriment, people feel obliged to extend such papers to include more, and often end up diluting the main message.
Recommendation
Publish (surpasses expectations and criteria for this Journal; among top 10%)
This is a terrific paper. It is short, simple, and beautiful. The result is clear and clearly presented. It should certainly be in SciPost.
We thank the referee for his interest in our manuscript and his support for publication.
I have a few minor comments. I found some typos. And I have a few suggestions.
Let me start with the minor things:
Q2.1 (1) Suggestion. Just after eq. 2.1, it is worth saying a few more words about why only neutral particles have a dispersion. I had to think for a while to understand what you meant and why this is true.
A2.1 We have added a sentence to the paper to highlight this important point.
Q2.2 (2) Change of language. I think about 7 lines up from the bottom of page 2, the word "should" would be better as "could", since in the very next sentence, you say this is not always the case.
A2.2 Thank you for the comment. We fixed the manuscript accordingly.
Q2.3 (3) Missing word. 6 lines below the start of the conclusion the word "is" is missing. (but that state IS annihilated)
A2.3 Thank you for the comment. We fixed the manuscript accordingly.
Now a larger suggestion for discussion of some additional phyiscs
which I think would add to the paper nicely:
Q2.4 I think it is worth comparing your result, at least for the case of the jain series, to the spectra expected from composite fermion excitons. Maybe this should be done in an appendix, maybe in the discussion section, or some combination. For nu=1/3, just by constructing all possible excitons --- a single particle in an excited CFLL and a single hole in the valence CF LL --- one obtains 1 state at L=1, 2 states at L=2, 3 states at L=3 and so forth in the thermodynamic limit. (This is trivial to show just by counting since the valence level has some angular momentum L0 and the first empty level has angular momentum L0+1 then next has L0+2 etc, and you want to add the hole in L0 to the electron in L0+M to generate some general L.) In the Jain picture, the L=1 exciton state vanishes under projection but it is unclear exactly why. In the HLR picture it is more clear (using some mass renormalized version of HLR --- see the reference from the other referee, thank you for the citation!). Here the Kohn mode is part of the spectrum, so it grabs one state at each L. So the low energy part of the spectrum then gets 0 states at L=1, 1 state at L=2, 2 states at L=3 and so forth. In your language would you say that we are seeing one spin-2 branch, one spin-3 branch and so forth?
A2.4 We thank the Referee for this very intriguing and intuitive suggestion. This said, we think that the counting needs to be considered more carefully. It is certainly correct that the L=1 belongs to the Kohn mode that the Referee suggested; however, after being projected to the lowest Landau level, some neutral excitations with the same L constructed in the composite fermion picture are not linearly dependent. For example, in a paper by Majumder, Mandal, and Jain. [Nature Phys. 5, 403 (2009)], some neutral excitations for $\nu =1 / 3$ constructed from the composite fermions are identical after projection to the lowest Landau level.
Q2.5 At $\nu=2/5$, just by counting excitons again, you get 1 state at L=1, 3 states at L=2, 5 states at L=3 and so forth. Again one mode of each L is pushed up to the Kohn mode leaving 0 states at L=1, 2 states at L=2, 4 states at L=3 and so forth. So you would say that we have 2 modes of spin 2, 2 more modes of spin 3 and so forth? Is this correct? If so, it then suggests that one should go looking for these in numerics. However, one thing that seems special about the magnetoroton is that, at least under some circumstances, it doesn't get buried in the continuum. Is there any hope that these higher spin modes could also be out of the continuum and could be observed?
A2.5 The neutral excitation counting for Jain’s state $\nu=2 / 5$ also needs to be considered more carefully. In a paper by Balram, Wójs, and Jain [Phys. Rev. B 88, 205312 (2013)], the counting of the neutral excitations in the composite fermion formalism has to take into account the linear dependence of neutral excitations with different constructions. For example, the neutral excitation with L=2 comes from the Lambda level $\Lambda=2$ to $\Lambda=4$ is excluded from the counting. If one uses the exclusion rules in [Phys. Rev. B 88, 205312 (2013)], then one can show that there are exactly 2(L-1) states with angular momentum L for (L \geq 2), constructed from the composite fermions. The counting of states with angular momentum L will fit with the construction of 2 neutral modes for each “spin” $s $ ($s \geq 2$). However, we do not know yet if the exclusion rules in [Phys. Rev. B 88, 205312 (2013)] are complete. So the count of neutral modes for each “spin” is still an open question for filling fraction $\nu= 2 / 5$.
For Jain states $ \nu=n/2n \pm 1 $, at the large n limit, we expect to have one spin-s state for $s \geq 2$, which are the deformation of the composite fermi-surface [Physical Review B 97 (19), 195314]. For the Jain state $\nu=n/4n \pm 1$, we expect one extra spin-2 mode [Phys. Rev. R 3 (3), 033217, Phys. Rev. Lett 128 (24), 246402].
Indeed, as the referee had already pointed out, the magneto-roton is special because it can be singled out from the continuum. The possibility of lowering the energy of the higher spin mode is still an open question.
We add some sentences discussing the implementation of our results with Jain's composite fermion construction in the conclusion section.
Q2.6 Because I was a bit slow in writing this report (just a few days!), I did have the benefit of reading one of the reports from another referee. That referee says that this work is "thin on results." I would actually disagree. While I would not object to extending this work or addressing some of the above issues (which overlaps with the issues raised by the other referee), I think the paper really does stand on its own. The paper makes a clear point and is an important point to make. It reminds me of the good old days of PRL (1950s) when a "letter" was often one page or less --- and these papers are very memorable. Now, much to our detriment, people feel obliged to extend such papers to include more, and often end up diluting the main message.
A2.6 We thank the referee again for the constructive comments and interesting suggestions.

---

## Round 1 · Referee Report · Anonymous (Referee 3) · 2025-6-26

Report
(*) The dispersion of the neutral excitation is assumed to be quadratic in Eq. (2.1). To what extent does the argument in Sec. 3 depend on this assumption? Is it possible that in the long wavelength limit some FQH neutral excitation modes do not obey the quadratic dispersion with k=0 as a maximum/minimum? I suggest the authors to discuss this issue. By the way, the effective mass in Eq. (2.1) can be negative, right? At least the magnetoroton mode of the \nu=1/3 Laughlin state in the lowest Landau level goes up when k->0, suggesting a negative mass.
(*) In the last paragraph of Sec. 2, the authors mentioned the difference between the low-energy excitations on the torus and on the sphere. This reminds me one thing which I didn’t understand well. For the \nu=1/3 Laughlin state, the magnetoroton mode on the sphere starts from angular momentum L=2 -- no states at L=1 and L=0 in this mode. So the long wavelength limit of this mode, the so called spin-2 graviton, should be carried by the L=2 state for finite systems. As the “momentum” on the sphere is often defined as L/R, where R is the sphere’s radius, this graviton actually has a finite momentum for finite systems on the sphere. As the Laughlin ground state has L=0, the graviton on the sphere has different L and (for finite systems) different “momentum” from the ground state. By contrast, spin-2 spectral function calculations on the torus, like those done in Refs. [5] and [6], found evidence of the spin-2 graviton exactly in the momentum k=0 sector even for finite systems. As the ground state is also in the k=0 sector on the torus, this means the graviton and the Laughlin ground state already have the same momentum for finite systems on the torus. This is quite different from the situation on the sphere. Why does not the graviton appear at a finite momentum (nonzero nx or ny) on the torus for finite systems? Can the authors comment on this discrepancy between different geometries?
Recommendation
Ask for minor revision

---

## Round 2 · Referee Report · Anonymous (Referee 3) · 2025-10-13

Report

All questions raised in my previous report have been answered satisfactorily. In particular I appreciate the authors' response to my question about the difference of graviton on the sphere and torus. I recommend the publication of the manuscript in SciPost Physics.

Recommendation

Publish (easily meets expectations and criteria for this Journal; among top 50%)

---

## Round 2 · Author Response

Dear Prof. Chris Laumann,

We would like to submit the revised version of our manuscript. We thank the referees for their comments/suggestions that help us strengthen our paper. We addressed in detail the comments/suggestions by the referees. We hope that we have answered all the queries of the three referees and that our work is now appropriate for publication in Scipost Physics.

Thank you for your consideration.

Best regards, Dung Xuan Nguyen and Dam Thanh Son

---

## Round 2 · List of Changes

Summary of changes:

-We fixed the grammar errors pointed out by the referee 2
-We have added a sentence to the paper to explain why our theory can only be applied to neutral excitations.
-We add some sentences discussing the implementation of our results with Jain's composite fermion construction in the conclusion section.
-We add a paragraph in Appendix A and footnote 2 to clarify that the choice of the dispersion relation doesn’t change the main conclusion of our paper.

---

## Editorial Decision

accepted_in_target_journal